# LONG-TAILED 3D DETECTION VIA 2D LATE FUSION

## ABSTRACT

Autonomous vehicles (AVs) must accurately detect objects from both common and rare classes for safe navigation, motivating the problem of Long-Tailed 3D Object Detection (LT3D). Contemporary LiDAR-based 3D detectors perform poorly on rare classes (e.g., CenterPoint achieves only 5.1 AP on `stroller`) because it is difficult to recognize objects from sparse LiDAR points alone. RGB images may help resolve such ambiguities, motivating the study of multi-modal RGB-LiDAR fusion. Specifically, we delve into a simple late-fusion framework that ensembles 2D RGB and 3D LiDAR detections and find that (a) high-resolution RGB images help recognize rare objects, (b) LiDAR provides precise 3D localization, and (c) uni-modal detectors can easily leverage more training data because they do not require aligning and annotating multi-modal data. We examine three critical components in this late-fusion framekwork: (1) whether to train 2D or 3D RGB detectors, (2) whether to match RGB and LiDAR detections in 3D or the projected 2D image plane (3) how to fuse matched detections. Extensive experiments reveal that using 2D RGB detectors, matching in the 2D image plane, and fusing scores probabilistically with calibration leads to the state-of-the-art LT3D performance, achieving 51.4 mAP on the established nuScenes LT3D benchmark, improving over prior work by 5.9 mAP.

## 1    INTRODUCTION

3D object detection is an integral component of the autonomous vehicle (AV) perception stack (Geiger et al., 2012; Caesar et al., 2020). To foster 3D perception research, the AV industry has released numerous large-scale multi-modal datasets (Caesar et al., 2020; Wilson et al., 2021; Sun et al., 2020). However, despite significant improvement in detecting common classes like `car` and `bus`, state-of-the-art detectors still perform poorly on rare classes like `stroller` and `debris`, which can impact downstream planning. This motivates the study of *Long-Tailed 3D Detection* (LT3D) (Peri et al., 2022).

**Status Quo**. Long-Tailed 3D Detection cannot be solved by simply training state-of-the-art (SOTA) detectors on both common and rare classes (Peri et al., 2022); e.g., BEVFusion (Liu et al., 2022) is a SOTA multi-modal transformer-based detector that achieves only 4.4 AP on the rare `child` class. Instead, Peri et al. (2022) finds that late-fusion of monocular 3D RGB (Wang et al., 2021) and LiDAR (Yin et al., 2021a) detections improves rare class recognition (cf. Fig. 1), achieving SOTA performance on the established nuScenes benchmark (Caesar et al., 2020). Importantly, Peri et al. (2022) show that (a) 3D LiDAR detectors achieve high recall but struggle to correctly recognize rare objects, and (b) RGB detectors are better at recognition but are unable to reliably estimate depth.

**Technical Insights.** To address LT3D, we delve into the simple late-fusion framework proposed by Peri et al. (2022) (cf. Fig. 1) and study three critical design choices (cf. Fig. 2), including whether to train 2D or 3D RGB detectors, where to match RGB and LiDAR detections (in the 2D image plane vs. 3D bird's-eye-view), and how to fuse matched detections.

First, we evaluate the impact of using 2D versus 3D RGB detectors for late-fusion, and find that the former is straightforward to train, can easily leverage external image data with 2D annotations, and leads to higher AP averaged over all classes. This is practically meaningful because annotating 2D boxes on RGB images is significantly cheaper than annotating 3D amodal cuboids.

Next, we study the impact of matching 2D RGB detections and 3D LiDAR detections on the 2D image plane versus in the 3D bird's-eye-view (BEV). Matching detections in the 3D BEV requires

Late-fusion requires *matching* and *fusing* uni-modal detections.

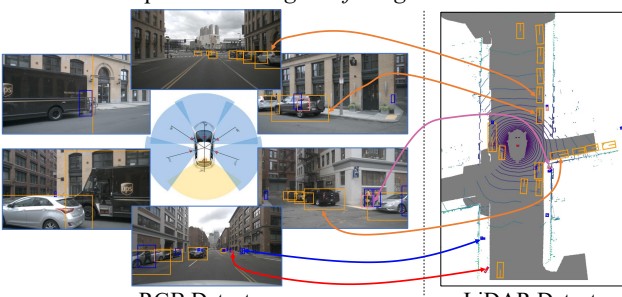

RGB Detector      LiDAR Detector

Figure 1: We explore a simple late-fusion framework for LT3D by ensembling RGB and LiDAR uni-modal detectors (Peri et al., 2022). We rigorously examine three critical components (cf. Fig. 2) and propose a simple method that fuses detections produced by a 2D RGB detector and a 3D LiDAR detector. Our method achieves 51.4 mAP on the LT3D benchmark based on the well-established nuScenes dataset (Caesar et al., 2020), significantly outperforming prior work (Liu et al., 2022) by 5.9 mAP (Table 1).

inflating 2D detections to 3D using depth imputed from LiDAR points (Wilson et al., 2020). This introduces additional depth estimation errors. Instead, we demonstrate that projecting 3D LiDAR detections to the 2D image plane for matching is more robust.

Lastly, we explore score calibration prior to fusion. We find that proper score calibration improves rare class detection and enables probabilistic fusion (Chiu et al., 2020) of LiDAR and RGB detections. Without score calibration, `rare`-class detections are often suppressed by overlapping `common`-class detections.

**Contributions**. We make three major contributions.

1. We extensively study three design choices within the late-fusion framework (cf. Fig. 2), and draw technical insights that generalize across different detector architectures.
2. Inspired by the above study, we present a simple late-fusion approach that effectively fuses 2D RGB-based detections and 3D LiDAR-based detections.
3. We conduct extensive experiments to ablate our design choices and demonstrate that our simple method achieves state-of-the-art results on LT3D benchmarks.

## 2    RELATED WORK

**3D Detection for Autonomous Vehicles (AVs)** can be broadly classified based on input modalities: LiDAR-only, RGB-only, and multi-modal detectors. Recent work in 3D detection is inspired by prior work in 2D detection (Zhou et al., 2020; Liu et al., 2016; Carion et al., 2020). LiDAR-based detectors like PointPillars (Lang et al., 2019), CBGS (Zhu et al., 2019), and PVRCNN++ (Shi et al., 2022) adopt an SSD-like architecture (Liu et al., 2016) that regresses amodal bounding boxes from a BEV feature map. More recently, CenterPoint (Yin et al., 2021a) adopts a center-regression loss that is inspired by Zhou et al. (2020). Despite significant progress, LiDAR-based detectors often produce many false positives because it is difficult to distinguish foreground objects from background solely on sparse LiDAR points. On the other hand, monocular RGB-based methods have gained increased interest in recent years due to low sensor cost and wide-spread adoption (Hu et al., 2023). FCOS3D (Wang et al., 2021) extends FCOS (Tian et al., 2019) by additionally regressing the size, depth, and rotation for each detection. Recent methods such as BEVDet and BEVFormer (Huang et al., 2021; Huang & Huang, 2022; Li et al., 2022c) construct a BEV feature-map by estimating per-pixel depth (Philion & Fidler, 2020). PolarFormer (Jiang et al., 2022b) introduces a polar-coordinate transformation that improves near-field detection. Importantly, many of these state-of-the-art 3D RGB detectors are commonly pre-trained on large external datasets like DDAD (Guizilini et al., 2020). Monocular RGB detectors accurately classify objects but struggle to estimate depth, particularly for far-field detections (Gupta et al., 2023). Despite recent advances in LiDAR and RGB 3D detectors, we find that multi-modal fusion is essential for LT3D (detailed next). Importantly, using both RGB (for better recognition) and LiDAR (for better 3D localization) helps detect rare classes. We delve into the late-fusion framework (cf. Fig. 1) to determine how to effectively fuse RGB and LiDAR uni-modal detectors for LT3D (cf. Fig. 2).

**Multimodal Fusion for 3D Object Detection** is an active area of research. Popular approaches can be categorized as input-fusion, feature-fusion, and late-fusion. Input-fusion methods typically augment LiDAR points using image-level features. For example, PointPainting (Vora et al., 2020) projects LiDAR points onto the output mask of a semantic segmentation model and appends corre-

Figure 2: We examine three key components in late-fusion of RGB and LiDAR uni-modal detectors: **A.** whether to train 2D or 3D monocular RGB detectors for late-fusion, **B.** whether to match uni-modal detections in the 2D image plane or 3D bird's-eye-view (BEV), and **C.** how to optimally fuse matched detections. Perhaps surprisingly, our exploration reveals that using 2D RGB detectors, matching in the 2D image plane, and fusing scores probabilistically with calibration leads to the state-of-the-art LT3D performance, significantly outperforming end-to-end trained multi-modal detectors (Table 1).

sponding class scores to each point. MVP (Yin et al., 2021b) densifies regions of LiDAR sweeps that correspond with objects in semantic segmentation masks. Frustum PointNets (Qi et al., 2018) leverages 2D RGB detections to detect objects within the frustum of the 2D RGB detection using PointNets (Qi et al., 2017). Recent works show that feature-fusion can be more effective than input-fusion. PointFusion (Xu et al., 2018) fuses global image and point-cloud features prior to detection and MSMDFusion (Jiao et al., 2022) fuses LiDAR and RGB features at multiple scales. TransFusion (Bai et al., 2022) and BEVFusion (Liu et al., 2022) fuse features in the BEV space using multi-headed attention. Despite the success of transformers for detecting common objects, Peri et al. (2022) find that TransFusion struggles to detect rare classes, as the transformer architecture, as adopted in TransFusion and BEVFusion, suffers from limited training data in the long tail. For transformers to work well in practice, they should be trained on diverse, large-scale datasets (Dosovitskiy et al., 2020; Radford et al., 2021; Li et al., 2022b). Further, end-to-end trained multi-modal detectors require paired multi-modal data for training. Therefore, we opt to study late fusion of uni-modal detectors, which do not require aligned RGB-LiDAR paired training data. CLOCs (Pang et al., 2020) is a late-fusion method that learns a separate network to fuse RGB and LiDAR detections, showing promising results for 3D detection. More recently, Peri et al. (2022) introduce a simple non-learned filtering algorithm that effectively removes false-positive LiDAR-detections that are far away from any 3D RGB detections. We delve into this simple (non-learned) late-fusion framework, study three crucial design choices, and present a method that significantly outperforms the state-of-the-art methods for LT3D.

**Long-Tailed Detection** is not unique to the AV domain and has been well studied in 2D (Gupta et al., 2019). Existing methods propose reweighting losses (Cui et al., 2019; Khan et al., 2017; Cao et al., 2019; Huang et al., 2019; Zhang et al., 2021), rebalancing data sampling (Drummond et al., 2003; Chawla et al., 2002; Han et al., 2005; Zhu et al., 2019), balancing gradients computed from imbalanced classes (Tang et al., 2020), and balancing network weights (Alshammari et al., 2022). CBGS (Zhu et al., 2019) explicitly addresses rare-class 3D detection by up-sampling LiDAR-sweeps with instances of rare classes, and pasting instances of rare objects copied from different scenes. Although it works well for improving the detection accuracy of uncommon classes (e.g., 5K~50K instances per class), it does not significantly improve for rare classes (e.g., <5K instances per class). Lee & Kim (2023) adopt data augmentation via sampling, and Jiang et al. (2022a) adopt active learning and hard example mining to obtain more data for rare classes. LT3D presents distinct opportunities and challenges compared to 2D long-tailed detection, because LiDAR sensors offer direct geometric and ego-motion cues that are difficult to extract from 2D images. Unlike 2D detectors, which must identify objects of various scales due to perspective image projection, 3D detectors do not experience as much scale variation for objects. However, LiDAR returns for far-field objects are sparse (Gupta et al., 2023), posing a different set of challenges. Additionally, rare classes, such as `child` and `stroller`, are typically small in size and have a limited number of LiDAR returns. As a result, LiDAR detectors struggle to accurately detect these rare classes. Our work addresses these issues by fusing RGB and LiDAR uni-modal detectors.

## 3 LATE-FUSION OF 2D-RGB AND 3D-LiDAR DETECTIONS FOR LT3D

As depicted in Fig. 1, our simple late-fusion framework ensembles uni-modal 2D RGB and 3D LiDAR detections. We first describe the benefits and drawbacks of using 2D detectors in Sec 3.1, present simple algorithms for matching 2D RGB and 3D LiDAR detections in Sec. 3.2, and finally describe score calibration and probabilistic fusion in Sec. 3.3.

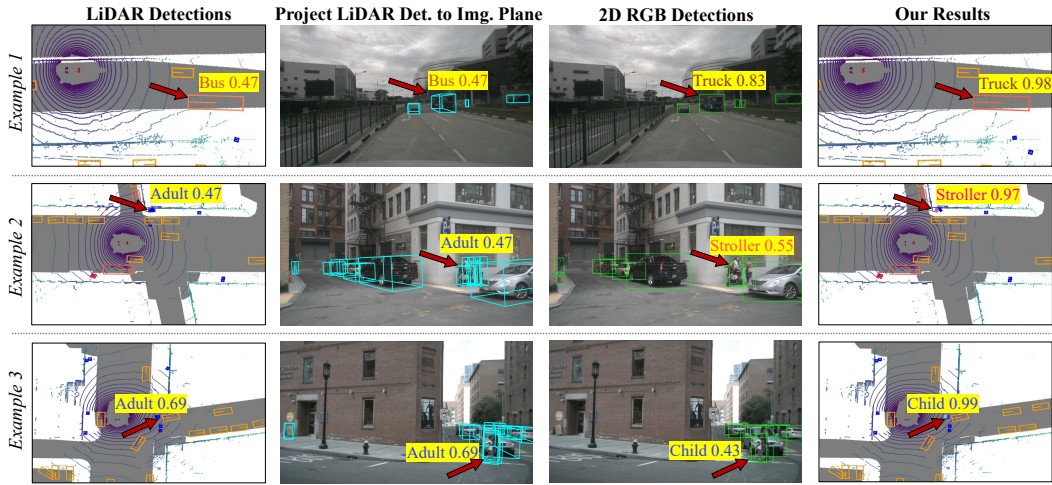

Figure 3: We highlight three examples to demonstrate how fusing 2D RGB and 3D LiDAR detections improves LT3D performance. In all examples, our method is able to correctly relabel detections which are geometrically similar (e.g., size and shape) in LiDAR but visually distinct in RGB, such as `bus-vs-truck`, `adult-vs-stroller`, and `adult-vs-child`. In addition, our method adopts probabilistic score fusion to boost confidence scores for matched 3D LiDAR and 2D RGB detections.

## 3.1 HOW DO WE INCORPORATE RGB INFORMATION, AND WHY?

Although LiDAR offers accurate localization, LiDAR-only detectors struggle to distinguish foreground objects from the background using sparse LiDAR alone. RGB images provide complementary information that is essential for identifying objects and disambiguating those that are geometrically similar in point clouds but semantically different in images. Although prior works address late-fusion, they combine *3D RGB detectors* with 3D LiDAR detections (Peri et al., 2022). In contrast, we find that *ensembling 2D RGB detectors* with 3D LiDAR detectors yields significantly better LT3D performance. We present insights on why using 2D detectors to incorporate RGB information yields better performance on rare categories below, and ablate the impact of using 2D vs. 3D RGB detectors for late-fusion in Table 2.

**2D RGB Detectors Are More Mature**. 2D object detection is a fundamental problem in computer vision (Felzenszwalb et al., 2009; Lin et al., 2014; Ren et al., 2015) that has matured in recent years and model trade-offs are well understood (Ren et al., 2015; Liu et al., 2016; Redmon et al., 2016; Lin et al., 2017). In this work, we consider two state-of-the-art 2D RGB detectors, YOLOV7 (Wang et al., 2022) and DINO (Zhang et al., 2022a). YOLOV7 is a real-time detector that identifies a number of training tricks that nearly doubles the inference efficiency over prior work without sacrificing performance. DINO is a recent transformer-based detector that improves upon DETR (Carion et al., 2020) using denoising anchor boxes. As 2D detectors do not make 3D predictions (e.g., depth and rotation), understanding how to best leverage them in the context of long-tailed 3D detection is a key challenge. We address this in subsection 3.2.

**2D RGB Detectors Can Train with More Diverse Data.** Training 2D RGB detectors only requires *2D bounding box* annotations, which are significantly cheaper to collect than 3D cuboids used for 3D RGB detector training (Wang et al., 2021; Jiang et al., 2022b). Since annotating 3D amodal cuboids is both expensive and non-trivial (compared to bounding-box annotations for 2D detection), datasets for monocular 3D RGB detection are considerably smaller and less diverse than their 2D detection counterparts. For example, nuScenes (published in 2020) annotates 144K RGB images of 23 classes (Caesar et al., 2020) while COCO (an early 2D detection dataset published in 2014) annotates 330K images (Lin et al., 2014) of 80 classes. As a result, large-scale 2D detection datasets are widely available. This allows pretraining 2D RGB detectors on significantly larger, diverse, publicly available datasets (Li et al., 2022a; Zhou et al., 2022b; Wang et al., 2019; Xu et al., 2020; Redmon & Farhadi, 2017). We demonstrate that using "freely available" 2D detection datasets helps train stronger 2D detectors, further improving LT3D performance (cf. Table 3).

## 3.2 How Do We Match Detections from Uni-Modal Detectors?

Finding correspondence between two sets of uni-modal detections is an essential step in the late-fusion framework (cf. Fig. 2**B**). Prior work (Peri et al., 2022) matches 3D RGB and 3D LiDAR detections using center distance in the bird-eye-view (BEV) plane. However, matching *2D RGB* and 3D LiDAR detections is non-trivial. Although prior work attempts to inflate 2D detections to 3D (using LiDAR points (Wilson et al., 2020)), we find that this procedure imputes additional noise, degrading match quality. Instead, we propose to match multi-modal detections by projecting 3D LiDAR detections to the 2D image plane. Importantly, this avoids adding additional noise due to imprecise depth estimates. We ablate the impact of matching in the 3D BEV vs. 2D image plane in Table 2, and present our 2D matching algorithm below.

**Spatial Matching in the 2D Image Plane**. Using the available sensor extrinsics, we project 3D LiDAR detections onto the 2D image plane (Pang et al., 2020). We then use the IoU metric to determine overlap between (projected) LiDAR and 2D RGB detections. A 2D RGB detection and a (projected) 3D LiDAR detection match if their IoU is greater than a fixed threshold. Although conceptually simple, it works better than using center distance to match detections in the 3D BEV (cf. Table 2).

**Handling Unmatched Detections**. Spatially matching multi-modal detections using 2D IU yields three categories of detections: matched detections, unmatched RGB detections, and unmatched LiDAR detections. We discuss how to fuse matched detections in the next subsection. For unmatched 2D RGB detections, we simply remove them. We posit that any unmatched RGB detections are likely to be false positives given that LiDAR detectors tend to yield high recall. On the other hand, we down-weight unmatched 3D LiDAR detections' confidence scores by multiplying by $w$. We set $w = 0.4$ in our work after tuning it via validation.

**Addressing Semantic Disagreement Between Modalities**. As illustrated by Fig. 2**C**, detections may match spatially but not semantically. To address this, we propose a semantic matching heuristic to better fuse LiDAR and RGB detections. Given a pair of spatially matched RGB and LiDAR detections, we consider two cases. If both modalities predict the same semantic class, we perform score-fusion (described next). Otherwise, if both modalities predict different semantic classes, we use the confidence score *and* class label of the RGB-based detection. Intuitively, RGB detectors can predict semantics more reliably from high resolution images. This helps correct misclassifications produced by the 3D LiDAR detector, as shown in Fig. 3.

## 3.3 How Do We Fuse Matched Detections?

We combine matched detections with the same semantic class produced by different uni-modal detectors using probablistic ensembling. However, the confidence scores of predictions from two detectors are not directly comparable. Therefore, we explore score calibration of RGB detections $x_{\text{RGB}}$ and LiDAR detections $x_{\text{LiDAR}}$. Score calibration is crucial to fairly compare scores for fusion.

**Score Calibration**. Score fusion requires the scores from independent uni-modal detectors to be comparable. In this work, we tune a temperature $\tau_c$ for the logit score of class-$c$ before applying a sigmoid transform (Guo et al., 2017; Chen et al., 2022), i.e., sigmoid(logit$_c/\tau_c$). Optimally tuning per-class $\tau_c$ is computationally expensive. Therefore, we greedily tune each $\tau_c$ for classes ordered by their cardinality via validation using the mAP metric.

**Probabilistic Ensembling**. Following Chen et al. (2022), we assume independent class prior $p(c)$, and conditional independence given the class label $c$, i.e., $p(x_{\text{RGB}}, x_{\text{LiDAR}}|c) = p(x_{\text{RGB}}|c)p(x_{\text{LiDAR}}|c)$. We compute the posterior/score (after calibration) $p(c|x_{\text{RGB}})$ and $p(c|x_{\text{LiDAR}})$. The final score is computed as

$$p(c|x_{\text{RGB}}, x_{\text{LiDAR}}) = p(x_{\text{RGB}}, x_{\text{LiDAR}}|c)p(c) \ / \ p(x_{\text{RGB}}, x_{\text{LiDAR}}) \quad \textit{\% Bayes Rule} \quad (1)$$
$$\propto p(x_{\text{RGB}}, x_{\text{LiDAR}}|c)p(c) \quad (2)$$
$$\propto p(x_{\text{RGB}}|c)p(x_{\text{LiDAR}}|c)p(c) \quad \textit{\% Conditional Ind.} \quad (3)$$
$$\propto p(c|x_{\text{RGB}})p(c|x_{\text{LiDAR}}) \ / \ p(c) \quad (4)$$

In the long-tailed scenario, priors $p(c)$ can have impact the final LT3D performance. To maximize performance, one needs to jointly tune all class priors $p(c)$. However, this is computationally expensive. Therefore, we tune them greedily, one by one ordered by class cardinality.

Table 1: **Comparison with the State-of-the-Art**. Recall that our late-fusion approach fuses 3D LiDAR and 2D RGB detections in the 2D image plane with score calibration and probabilistic ensembling. It performs the best on *all* categories, notably improving over the end-to-end multi-modal BEVFusion (Liu et al., 2022) for classes with `medium` examples by 7.4% and `few` examples by 7.2%.

| Method | Multi-Modal | All | Many | Medium | Few |
|---|---|---|---|---|---|
| FCOS3D (RGB-only) (Wang et al., 2021) | | 20.9 | 39.0 | 23.3 | 2.9 |
| BEVFormer (RGB-only) (Li et al., 2022c) | | 27.3 | 52.3 | 31.6 | 1.4 |
| PolarFormer (RGB-only) (Jiang et al., 2022b) | | 28.0 | 54.0 | 31.6 | 2.2 |
| CenterPoint (LiDAR-only) (Yin et al., 2021a) | | 39.2 | 76.4 | 43.1 | 3.5 |
| TransFusion-L (LiDAR-only) (Bai et al., 2022) | | 38.5 | 68.5 | 42.8 | 8.4 |
| BEVFusion-L (LiDAR-only) (Liu et al., 2022) | | 42.5 | 72.5 | 48.0 | 10.6 |
| CMT-L (LiDAR-only) (Yan et al., 2023) | | 34.7 | 73.4 | 35.9 | 1.1 |
| TransFusion (LiDAR + RGB) (Bai et al., 2022) | ✓ | 39.8 | 73.9 | 41.2 | 9.8 |
| BEVFusion (LiDAR + RGB) (Liu et al., 2022) | ✓ | 45.5 | 75.5 | 52.0 | 12.8 |
| DeepInteraction (LiDAR + RGB) (Yang et al., 2022) | ✓ | 43.7 | 76.2 | 51.1 | 7.9 |
| CMT (LiDAR + RGB) (Yan et al., 2023) | ✓ | 44.4 | **79.9** | 53.0 | 4.8 |
| CenterPoint + FCOS3D (Peri et al., 2022) | ✓ | 43.6 | 77.1 | 49.0 | 9.4 |
| **Ours** | ✓ | **51.4** | 77.9 | **59.4** | **20.0** |

## 4 EXPERIMENTS

In this section, we conduct extensive experiments to evaluate our proposed approach. We compare our late-fusion approach with prior works and present a detailed ablation study that further addresses the three motivating questions in Fig. 2. We find that our proposed approach improves over prior works by 5.9% averaged over all classes, notably improving by 7.2% on rare classes. We evaluate our method on the Argoverse 2 dataset (Wilson et al., 2023) in the appendix and find that the same conclusions hold.

### 4.1 EXPERIMENT SETUP

**Datasets**. We study LT3D using the well-established nuScenes dataset (Caesar et al., 2020). We follow the protocol defined by Peri et al. (2022), using all the 18 long-tail categories. Moreover, we use the nuImages dataset (Caesar et al., 2020) as an external 2D annotated data source to study how using additional data to train better 2D RGB detectors improves late-fusion performance.

**Metrics**. Mean average precision (mAP) is an established metric for object detection (Lin et al., 2014). For 3D detection, a true positive (TP) is defined as a detection that has a center distance within a distance threshold on the ground-plane to a ground-truth annotation (Caesar et al., 2020). mAP computes the mean of AP over classes, where per-class AP is the area under the precision-recall curve drawn with distance thresholds of [0.5, 1, 2, 4] meters. We report the metrics for three groups of classes based on their cardinality: Many (>50k instances per class), Medium (5k~50k), and Few (<5k). Following Peri et al. (2022), we use the official nuScenes train-set for training and evaluate on its val-set.

### 4.2 COMPARISON TO STATE-OF-THE-ART

We compare our late-fusion approach against prior works in Table 1, and present qualitative results in Fig. 3. We adapt existing methods (which were previously trained on the standard 10 class in the official nuScenes benchmark) for LT3D by retraining all them on all the 18 classes. We provide additional implementation details in the appendix.

CenterPoint (Yin et al., 2021a), a popular 3D LiDAR detector, is unable to detect rare objects, achieving just 3.5% AP on classes with `few` examples. This is expected as it is difficult to identify rare objects from sparse LiDAR points alone. Perhaps surprisingly, the transformer-based 3D LiDAR detector BEVFusion-L performs considerably better on rare classes, achieving 10.6 % AP. However, BEVFusion-L performs 3.9% worse than CenterPoint on common classes. We posit that it is difficult to learn transformer models on rare classes that do not have sufficient amount of data, and the long-tail issue exacerbates training.

Table 2: **Fusing Detections in the 3D BEV vs. 2D Image Plane**. We evaluate the impact of fusing 3D LiDAR detections (from CenterPoint trained with hierarchical loss (Peri et al., 2022)) with 2D RGB and 3D RGB detections in both the 3D BEV and 2D image plane. We match and filter detections in the 3D BEV using center distance as proposed by Peri et al. (2022). In contrast, we match and filter detections in the 2D image plane using IoU. Following Wilson et al. (2020), we inflate 2D detections to the 3D BEV using LiDAR points within the box frustum. We project 3D detections to the 2D plane using provided sensor extrinsics. We find that matching 3D RGB detections in the 3D BEV and in the 2D image plane yields similar results. Unsurprisingly, inflating 2D RGB detections for matching in the 3D BEV performs worse than matching 3D RGB detections in the 3D BEV. In contrast, filtering LiDAR-based detections using 2D detections in the 2D image plane (bottom right panel) significantly improves performance for classes in medium and few by >10 mAP. This suggests that 2D detectors achieve better detection performance compared to 3D RGB detectors.

| Method | Fusion in 3D BEV | | | | Fusion on 2D Image Plane | | | |
|---|---|---|---|---|---|---|---|---|
| | All | Many | Medium | Few | All | Many | Medium | Few |
| CenterPoint | 40.8 | 76.5 | 45.3 | 5.7 | 40.8 | 76.5 | 45.3 | 5.7 |
| + FCOS3D (Wang et al., 2021) | 42.9 | 76.6 | 48.7 | 8.1 | 42.6 | 75.0 | 49.4 | 7.7 |
| + BEVFormer (Li et al., 2022c) | 43.2 | 76.9 | 50.8 | 6.3 | 42.8 | 75.2 | 51.4 | 5.7 |
| + PolarFormer (Jiang et al., 2022b) | 42.8 | 76.8 | 50.0 | 6.1 | 42.6 | 75.1 | 51.1 | 5.6 |
| + YOLOV7 (Wang et al., 2022) | 40.1 | 76.1 | 43.8 | 5.8 | 45.7 | 77.1 | 52.8 | 11.2 |
| + DINO (Zhang et al., 2022a) | 40.3 | 76.2 | 44.1 | 5.9 | 49.5 | 77.4 | 57.7 | 16.7 |

In contrast, BEVFusion (Li et al., 2022c), which is an end-to-end trained multi-modal method, performs better than the LiDAR-only variant (BEVFusion-L), confirming the benefit of using multi-modal input for LT3D. Peri et al. (2022) introduce a simple filtering algorithm that keeps CenterPoint detections that are close (based on center distance) to monocular 3D RGB detections produced by FCOS3D in the BEV, and discards all other LiDAR predictions. Given the success of this simple approach, we study this late-fusion paradigm further. By carefully considering design choices outlined in Fig. 2, we improve over (Peri et al., 2022) by 7.8%.

## 4.3 ABLATION STUDY

We design a set of experiments to study the trade-off between using 2D and 3D RGB detectors, and matching in the 2D image and 3D BEV plane (cf. Table 2). Further, we examine the impact of using additional data and study different fusion strategies (cf. Table 3). Our analysis confirms that 2D RGB are better suited for late-fusion, matching projected 3D LiDAR detections in the 2D image-plane outperforms matching 3D RGB detections in the 3D BEV, and score calibration prior to probabilistic fusion improves performance.

**How Do We Incorporate RGB Information, and Why?** Although LiDAR-based detectors are widely adopted for 3D detection, they produce many high-scoring false positives (FPs) for rare classes due to misclassification (Peri et al., 2022). We focus on removing such FPs using an RGB-based detector. We leverage two insights: (1) LiDAR-based 3D detections are accurate w.r.t 3D localization and yield high recall (though classification is poor), and (2) RGB-based detections are accurate w.r.t recognition (though 3D localization is poor). We incorporate RGB information by matching and filtering 3D LiDAR detections with RGB-based detections. We evaluate the impact of using 2D RGB-based detectors (e.g., YOLOV7 and DINO) vs. 3D RGB-based detectors (e.g., FCOS3D, BEVFormer, PolarFormer) in Table 2.

In this work, we consider the impact of matching LiDAR detections with 3D RGB detections in the 3D BEV and 2D image plane. Similarly, we consider the impact of matching LiDAR detections with 2D RGB detections in the 3D BEV and 2D image plane.

**How Do We Match Detections From Uni-Modal Detectors?** To match 3D detections in the 2D image plane we can use the provided sensor extrinsics. To match 2D detections in the 3D BEV, we can inflate the 2D detection using LiDAR points within the box frustum. In practice, we find that naively lifting 2D RGB detections into 3D leads to imprecise depth estimates and incorrect matches.

To match and filter LiDAR and RGB detections in the 3D BEV, we follow the procedure prescribed by Peri et al. (2022). For each RGB-based detection, we keep LiDAR-based detections within a radius of $m$ meters and remove all the others (that are not close to any RGB-based detections). This works well for 3D RGB detections. FCOS3D, BEVFormer, and PolarFormer improve over the LiDAR-only model by 2% averaged over all classes (cf. Table 2). Unsurprisingly, we find that

Table 3: **Ablation on Multi-Modal Fusion**. Our analysis confirms that 2D RGB is better suited for late-fusion, matching projected 3D LiDAR detections in the 2D image-plane outperforms matching 2D RGB detections inflated to the 3D BEV, and score calibration prior to probabilistic fusion improves performance.

| Method | All | Many | Medium | Few |
|---|---|---|---|---|
| CenterPoint w/ Hierarchy (Peri et al., 2022) | 40.4 | 77.1 | 45.1 | 4.3 |
| + 2D Img. Filtering w/ DINO | 47.9 | 77.1 | 55.8 | 14.4 |
| + External Data | 49.8 | 77.1 | 57.1 | 18.6 |
| + Score Calibration | 50.9 | 77.9 | 59.2 | 18.7 |
| + Probabilistic Ensembling | **51.4** | **77.9** | **59.4** | **20.0** |

inflating 2D RGB detections for matching in the 3D BEV performs worse than matching 3D RGB detections in the 3D BEV. YOLOV7 and DINO both perform worse than the LiDAR-only baseline. To match and filter LiDAR and RGB detections in the 2D image plane, we simply use the IoU metric. Two detections are considered matched if their spatial overlap exceeds a fixed threshold. We find that projecting LiDAR detections in the 2D image plane and filtering them using 2D RGB detections significantly improves performance for classes with `medium` and `few` examples by more than 10%. Surprisingly, we find that projecting 3D RGB detections for matching in the 2D image plane performs worse than matching 2D RGB detections in the 2D image plane, suggesting that 2D detectors achieve better recognition performance compared to 3D RGB detectors. We emperically verify this claim in the appendix.

**How Do We Fuse Matched Detections?** Prior to fusion, we first calibrate the scores of LiDAR and RGB detections to ensure that they are comparable. This improves accuracy by 1% averaged over all classes, notably improving performance for classes with a `medium` number of examples. When using probabilistic ensembling, we use Bayesian fusion to reason about the final score of overlapping detections. Concretely, if two matched detections fire in the same place, the fused score should be higher than the individual scores because there is twice the evidence of an object at that particular spatial location. As shown in Table 3, probabilistic ensembling further improves accuracy by 0.6 mAP.

**Per-Class Breakdown Results.** We highlight the per-class performance of recent multi-modal methods in Table 4. All multi-modal methods perform similarly on `common` classes. However, we find that all multi-modal methods perform considerably worse on classes in-the-tail compared to common classes, highlighting the need for further investigation by the research community. Notably, our late-fusion approach achieves 20% higher AP on `pushable-pullable` and 6% higher AP on `stroller`. In general, our late-fusion approach yields considerable improvement on classes with `medium` and `few` examples. Despite significant improvements in rare class detection accuracy, our approach detects `child` with 8% AP. We posit that it is difficult to distinguish `child` from `adult` due to perspective geometry, since it is difficult to distinguish a small child close to the camera vs. a tall adult far away from the camera.

**Failure Cases and Visualizations**. We visualize common failure cases of our late-fusion approach and compare them with the failure cases of TransFusion, an end-to-end trained multi-modal detector. We find that our method fails in cases of occlusions (where there is no 3D information) and in cases where the 2D RGB detector misclassifies the object. See Fig. 4 for detailed analysis.

Table 4: **nuScenes Per-Class Breakdown.** Our late-fusion approach achieves the highest per-class AP on 6 out of 10 classes shown below. Out late-fusion approach significantly improves over DeepInteraction, improving `bicycle` accuracy by 5.8%, `construction worker` by 15.2 %, `stroller` by 6.8 %, and `pushable-pullable` by 17.3 %. Note, CV is `construction vehicle`, MC is `motorcycle`, PP is `pushable-pullable`, CW is `construction-worker`, and Stro. is `stroller`. We highlight classes with `Medium` and `Few` examples per class in blue.

| Method | Car | Adult | Truck | CV | Bicy | MC | Child | CW | Stro. | PP |
|---|---|---|---|---|---|---|---|---|---|---|
| TransFusion (Bai et al., 2022) | 84.4 | 84.2 | 58.4 | 24.5 | 46.7 | 60.8 | 3.1 | 21.6 | 13.3 | 25.3 |
| BEVFusion (Liu et al., 2022) | **90.2** | 67.2 | **65.5** | 35.2 | 58.8 | 77.0 | 4.4 | 39.0 | 29.6 | 34.1 |
| DeepInteraction (Yang et al., 2022) | 84.9 | 85.9 | 63.2 | 35.3 | 64.3 | 76.2 | 6.0 | 30.7 | 30.9 | 30.8 |
| CMT (Yan et al., 2023) | 88.6 | **87.7** | 65.2 | **36.9** | 66.7 | **76.3** | 4.7 | 34.4 | 9.4 | 34.1 |
| CenterPoint + FCOS3D (Peri et al., 2022) | 88.5 | 86.6 | 63.4 | 29.0 | 58.5 | 68.2 | 5.3 | 35.8 | 31.6 | 39.3 |
| **Ours** | 86.3 | 86.2 | 60.6 | 35.3 | **70.1** | 75.9 | **8.8** | **55.9** | **37.7** | **58.1** |

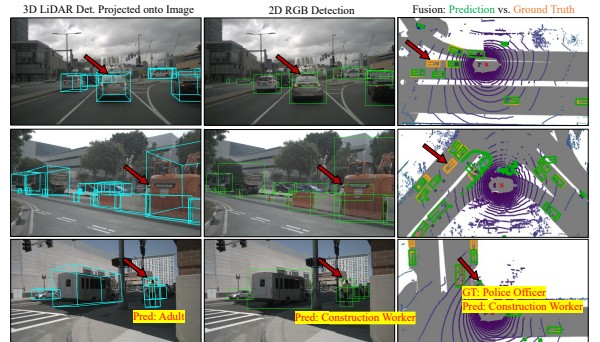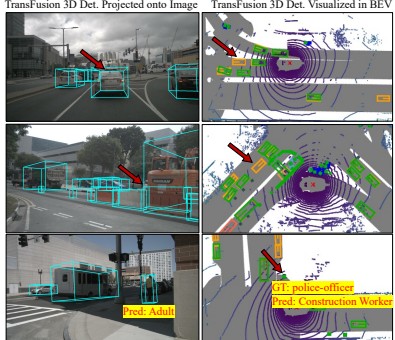

Figure 4: **Failure Cases**. Both our method (columns 1-3) and TransFusion Bai et al. (2022) (columns 4 -5) have the same failure cases. In the first and second row, the 2D RGB detector DINO detects the heavily occluded cars but 3D LiDAR detector fails to detect them. As a result, the late-fusion predictions miss these cars because our method throws away unmatched RGB-detections for which we do not have accurate 3D information. In the third row, we see that although both the LiDAR and RGB detectors fire on the object (whose ground-truth label is `police-officer`), LiDAR detector predicts it as `adult` and RGB detector predicts it as `construction-worker`. As a result, the final detection is incorrect w.r.t the predicted categorical label. TransFusion also misclassifies this object and predicts it as an `adult`.

## 5 CONCLUSION

We present a detailed exploration of late-fusion, focused on addressing three key design decisions. We find that 2D RGB detectors are better suited for late-fusion, matching projected 3D LiDAR detections in the 2D image-plane outperforms matching 2D RGB detections inflated to the 3D BEV, and score claibration prior to probabilistic fusion improves performance. Our simple late-fusion approach achieves state-of-the-art performance, improving over prior work by 5.9% mAP. Despite the success of transformers in other computer vision and natural language doamins, we find that end-to-end transformer-based multi-modal detectors still struggle to detect rare classes.

**Limitations**. Our work focuses on LT3D, a problem that emphasizes 3D object detection for rare classes which are often safety-critical for AVs, like `stroller` and `debris`. Therefore, improving LT3D is important for ensuring safe autonomy. However, our work does not directly study how addressing LT3D affects downstream perception tasks. Future work should address this limitation. While our late-fusion pipeline can fuse detections from *any* detectors, we focus on fusing only detections from uni-modal detectors. We hope future work will study fusing detections from more uni- and multi-modal detectors.

**Future Work.** As shown in Fig. 5, simply training better 2D RGB detectors with more data yields a natural pathway for improving LT3D performance. We find that 2D detection accuracy on nuScenes is a strong proxy for final 3D LT3D performance. Recent work in large-scale vision language models (Zhou et al., 2022a; Zhang et al., 2022b; Minderer et al., 2023) show promising zero-shot results in detecting rare classes. Identifying ways of incorporating foundation models into our late-fusion framework can greatly improve LT3D.

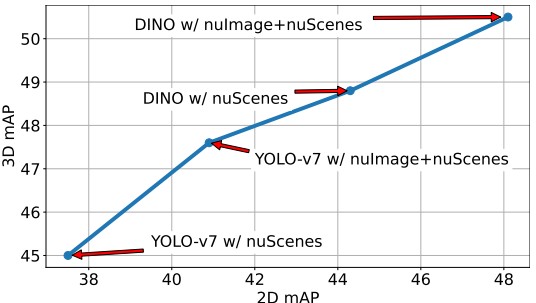

Figure 5: Although nuScenes is a 3D detection benchmark, we can generate 2D annotations using the provided sensor extrinsics by projecting the 3D annotations to the 2D image plane. We find that evaluating 2D detectors using these 2D nuScenes annotations is a good proxy task (x-axis) that is positively correlated with the downstream performance of the full late-fusion pipeline (y-axis). Concretely, training 2D detectors with more data (e.g. training with nuScenes and nuImages), and using stronger 2D detectors (e.g. DINO) improves performance on the proxy task as well as the downstream late-fusion algorithm.

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

# Appendix

## A    IMPLEMENTATION DETAILS

We retrain several existing LiDAR-only, RGB-only, and multi-modal detectors for LT3D by simply increasing the number of target classes from 10 (standard nuScenes setup) to 18 (LT3D nuScenes setup). We employ all standard augmentation techniques, including copy-paste augmentation and adopt the sampling ratios defined by Peri et al. (2022) for all detectors. Following the standard nuScenes setup, we aggregate the past 10-frames for LiDAR densification. We assume that we are provided with ego-vehicle poses for prior frames to align all LiDAR sweeps to the current ego-vehicle pose. Since LiDAR returns are sparse, this densification step is essential for accurate 3D detection.

We use the open source implementations from Peri et al. (2022) for training FCOS3D, CenterPoint, and TransFusion and use the first-party implementation of all other detectors. We follow the training schedule proposed by each respective paper. We train our model with 8 RTX 3090 GPUs. The training noise (from random seed and system scheduling) is < 1% of the accuracy (standard deviation normalized by the mean).

By default, we train the 2D RGB on the 2D bounding boxes derived from 3D annotations from nuScenes and additionally train with 2D bounding boxes from nuImages where denoted. Our 2D RGB detectors, YOLOV7 and DINO, are pre-trained on the ImageNet (Deng et al., 2009) and COCO (Lin et al., 2014) datasets. We provide our code in the supplement and will make our code public to support future exploration into LT3D.

## B    TRANSFORMER-BASED LIDAR DETECTORS

In the main paper, we evaluate all late-fusion methods by filtering LiDAR detections from CenterPoint with different RGB detectors. However, our late-fusion approach generalizes to other 3D LiDAR detectors. To this end, we evaluate our best-performing late-fusion configuration from the main paper with LiDAR detections from TransFusion-L (Bai et al., 2022) (cf. Table 5). First, we note that TransFusion-L performs 3.7% worse than CenterPoint on long-tailed 3D detection, confirming the results from Peri et al. (2022). Specifically, TransFusion has lower recall across all classes compared to CenterPoint. We posit that this lower recall is due to its limited number of queries (200 by default). Additionally, unlike traditional CNN-based detectors that can predict multiple boxes with different class names and confidence scores, TransFusion cannot get partial credit by drawing multiple boxes with different class names around the same object by because the bipartite matching loss strongly penalizes this behavior. Despite its lower performance, we find that filtering TransFusion's 3D detections with DINO's 2D detections in the image plane yields a 5% improvement overall, and a 7% AP improvement for rare classes. In contrast, the end-to-end multi-modal variant of TransFusion only improves over the LiDAR-only baseline by 1% overall.

Table 5: **Transformer-based 3D LiDAR Detector**. We evaluate the late-fusion performance of TransFusion-L (LiDAR-only) with YOLOV7 and DINO, and find that our late fusion strategy is able to provide consistent improvement over the LiDAR-only model, notably improving by 5% overall, and providing a 7% AP improvement for rare classes.

| Method | All | Many | Medium | Few |
|---|---|---|---|---|
| CenterPoint *w/ Hier.* (Peri et al., 2022) | 40.8 | 76.5 | 45.3 | 5.7 |
| + 2D Img. Filtering w/ YOLOV7 | 47.6 | 77.4 | 54.6 | 14.6 |
| + 2D Img. Filtering w/ DINO | **51.4** | **77.9** | **59.4** | **20.0** |
| TransFusion-L *w/ Hier.* | 37.1 | 69.8 | 40.8 | 5.4 |
| + 2D Img. Filtering w/ YOLOV7 | 40.0 | 69.9 | 47.7 | 6.2 |
| + 2D Img. Filtering w/ DINO | 42.5 | 69.9 | 48.7 | 12.5 |
| TransFusion (RGB + LiDAR) | 38.2 | 70.6 | 42.6 | 6.1 |

## C  2D DETECTION RESULTS

We evaluate the 2D detection performance of state-of-the-art 2D and 3D RGB detectors on nuScenes to demonstrate that simply projecting 3D detections to the 2D image plane yields poor late-fusion results. Recall, we project all 3D bounding boxes to 2D using the provided sensor extrinsics for training and evaluation. All detectors are only trained on the nuScenes train-set. Table 6 shows that 2D detectors outperform state-of-the-art 3D RGB detectors, e.g., BEVFormer (ECCV'22) for 2D detection on the nuScenes val-set. Importantly, DINO performs significantly better than BEVFormer (15.9 vs. 2.1 mAP) on classes with `few` examples.

Table 6: We find that simply projecting 3D detections from RGB-only detectors to 2D yields considerably lower 2D detection accuracy across all classes. Instead, we find that 2D detectors achieve higher performance across all cardinalities. This explains why fusing 2D detections from 2D detectors in the 2D image plane yields the best results.

| Method | All | Many | Medium | Few |
|---|---|---|---|---|
| FCOS3D (**3D** RGB Detector ) | 18.3 | 36.0 | 21.1 | 0.2 |
| PolarFormer (**3D** RGB Detector) | 20.7 | 37.5 | 25.2 | 1.6 |
| BEVFormer (**3D** RGB Detector) | 23.0 | 40.8 | 28.2 | 2.1 |
| YOLOV7 (**2D** RGB Detector) | 37.5 | 63.5 | 45.0 | 7.1 |
| DINO (**2D** RGB Detector) | **44.3** | **67.8** | **51.9** | **15.9** |

## D  INFERENCE TIMING RESULTS

We compare the inference runtime of our method and compare against prior work on a single A100 GPU (with a batch size of 1) following the protocol described in Peri et al. (2023). As uni-modal RGB and LiDAR detectors run in parallel, the overhead of score calibration and fusion is negligible. As a result, our fusion method (last column) has the same runtime as CenterPoint (CP), and is faster than TransFusion (Bai et al., 2022), BEVFormer (Li et al., 2022c) and DeepInteraction (Yang et al., 2022).

Table 7: **Inference Timing Results**. We find that our proposed approach is often faster than existing state-of-the-art methods. Since our late-fusion approach requires running two uni-modal detectors, we can speed up inference by running these in parallel. Notably, our fusion approach is light-weight and requires minimal overhead at inference.

| | FCOS3D | BEVFormer | CP | TransFusion | DeepInteraction | CP+FCOS3D | **Ours** |
|---|---|---|---|---|---|---|---|
| | Wang et al. (2021) | Li et al. (2022c) | Yin et al. (2021a) | Bai et al. (2022) | Yang et al. (2022) | Peri et al. (2022) | |
| Speed (ms) | 89 | 327 | 323 | 367 | 590 | 324 | 323 |

## E  AP VS. NDS RESULTS

Although we report mAP following Peri et al. (2022), we compare methods w.r.t nuScenes Detection Score (nDS) and mAP, an find that methods follow the same ranking on both metrics. Since NDS is computed as a weighted sum of mAP and other true positive metrics, with mAP weighted five times greater than other components, it is unsurprising that all trends from the main paper hold.

Table 8: **NDS Results**. We compare methods reported in the main paper w.r.t nuScenes Detection Score (NDS) and mAP. We find that methods follow the same rankings on both metrics.

| | FCOS3D | BEVFormer | CP | TransFusion | DeepInteraction | CP+FCOS3D | **Ours** |
|---|---|---|---|---|---|---|---|
| | Wang et al. (2021) | Li et al. (2022c) | Yin et al. (2021a) | Bai et al. (2022) | Yang et al. (2022) | Peri et al. (2022) | |
| mAP | 20.9 | 27.3 | 39.2 | 39.8 | 43.7 | 43.6 | 51.4 |
| NDS | 30.4 | 38.8 | 54.9 | 53.9 | 54.4 | 56.7 | 60.4 |

Table 9: **Comparison with the Argoverse 2 State-of-the-Art**. We present results AV2 evaluated at 50 meters. FCOS3D achieves poor performance, likely due to inaccurate depth estimates. In contrast, CenterPoint achieves strong performance on all classes. Our multi-modal fusion approach significantly improves over CenterPoint, achiving 8.3% improvement averaged over all classes. These results on AV2 are consistent with those on nuScenes (cf. Table 1), demonstrating the general applicability of our approach.

| Method | Multimodal | All | Many | Medium | Few |
|---|---|---|---|---|---|
| FCOS3D Wang et al. (2021) (RGB-only) | | 14.6 | 27.4 | 17.0 | 7.8 |
| CenterPoint Yin et al. (2021a) (LiDAR-only) | | 44.0 | 77.4 | 46.9 | 30.2 |
| TransFusion-L Bai et al. (2022) (LiDAR-only) | | 44.0 | 77.4 | 46.9 | 30.2 |
| CenterPoint + FCOS3D Peri et al. (2022) | ✓ | 48.4 | 79.0 | 51.4 | 35.3 |
| **Multi-Modal Late Fusion (Ours)** | ✓ | **52.3** | **89.4** | **54.2** | **38.7** |

# F   ARGOVERSE 2 RESULTS

We present results on the large-scale Argoverse 2 (AV2) dataset, another long-tailed dataset developed for autonomous vehicle research. AV2 evaluates on 26 classes, which follow a long-tailed distribution. Following Peri et al. (2022), we train on and evaluate detections up to 50m. As show in Table 9, our main conclusions from nuScenes still hold for AV2. FCOS3D yields poor performance on all classes, likely due to inaccurate depth estimates. CenterPoint performs considerably better, achieving high accuracy on classes with `many` examples. Notably, CenterPoint performs better on AV2's rare classes (30.2 AP) compared to nuScenes's rare classes (3.5 AP), likely because AV2 has more examples per-class in-the-tail. Lastly, our proposed late-fusion approach yields an 8.3% improvement overall, improving performance for classes of all cardinalities. These new results on AV2 are consistent with those on nuScenes, demonstrating the general applicability of our approach.

