# OpenReview forum: "Long-Tailed 3D Detection via 2D Late Fusion"
_ICLR.cc/2024/Conference — ICLR 2024 Conference Withdrawn Submission_

### Official Review · Reviewer_ezvM · 2023-10-13

**Soundness:** 2 fair
**Presentation:** 3 good
**Contribution:** 1 poor
**Rating:** 3
**Confidence:** 5

**Summary:**

This paper propose to late-fuse 2D RGB detectors and 3D LiDAR detections in the 2D image-plane. Experiments on  the established nuScenes LT3D benchmark show promising results.

**Strengths:**

Experiments on  the established nuScenes LT3D benchmark show promising results.

**Weaknesses:**

see questions part.

**Questions:**

I am still surprised that this late fusion method can achieve such a big improvement in indicators. In fact, in addition to the Frustum PointNets you mentioned, articles such as SRDL[1] , MMF[2] also propose ways to integrate 2D detection results into 3D.

In terms of results, the original intention of this article is to solve the detection problem of rare categories, but there is no emphasis in this direction from the experimental level. From the perspective of the idea, this paper is more like a fusion of 2D and 3D detection results, so from the perspective of contribution, it seems a bit weak.

And the experimental results of this paper were done on the nuScenes data set, so how does it perform on the KITTI data set and the larger Waymo data set?

Finally, the first sentence of the second paragraph of Section 3.2 seems a bit ambiguous and seems to violate the principle of anonymity.



[1] He Q, Wang Z, Zeng H, et al. Stereo RGB and Deeper LIDAR-Based Network for 3D Object Detection in Autonomous Driving[J]. IEEE Transactions on Intelligent Transportation Systems, 2022, 24(1): 152-162.

[2] M. Liang, B. Yang, Y. Chen, R. Hu, and R. Urtasun, “Multi-task multi-sensor fusion for 3D object detection,” in Proc. IEEE/CVF Conf.
Comput. Vis. Pattern Recognit., Jun. 2019, pp. 7345–7353

---

### Official Review · Reviewer_zAdM · 2023-10-27

**Soundness:** 1 poor
**Presentation:** 2 fair
**Contribution:** 2 fair
**Rating:** 5
**Confidence:** 4

**Summary:**

In this paper, the late-fusion of LiDAR and camera detection results is investigated for the long-tailed distribution problem in 3D object detection. By examining the different choices of 2D/3D RGB detectors, the choice of geometric fusion planes for LiDAR and image detection results, and the probabilistic fusion of LiDAR-camera multi-modal results, a significant improvement of the detection performance of long-tailed distribution 3D objects on the challenging dataset nuScenes is achieved. However, as far as the content of the article is concerned, I have the following questions and suggestions:
1) The article is not innovative enough (although the results are impressive) and gives the impression of reading an experimental report rather than a scientific paper.
2) The motivation of the fusion strategies used is not clearly described in the article.
3) How to greedily tune the temperatures and the priors are not clear.
4) Whether the way of late-fusion is limited by the upper limit of the capacity of the uni-modal detector itself. Would the conclusion be different if a different backbone is used?
5) The fusion choice in the image plane requires LiDAR and camera to have sufficient field of view overlap. In the experiments, are the detection results of images from all viewpoints fused? If yes, how to solve the inconsistency of the detection results of images from different viewpoints in the overlap area?

**Strengths:**

1)	The overall detection results on nuScenes are impressive.
2)	The ablation studies verify the effectiveness of each choice and design.

**Weaknesses:**

1) The article is not innovative enough and gives the impression of reading an experimental report rather than a scientific paper.
2) The motivation of the fusion strategy used is not clearly described in the article.
3) How to greedily tune temperatures and the priors are not clear.

**Questions:**

1) Whether the way of late-fusion is limited by the upper limit of the capacity of the uni-modal detector itself. Would the conclusion be different if a different backbone is used?
2) The fusion method in the image plane requires LiDAR and camera to have sufficient field of view overlap. In the experiments, are the detection results of images from all viewpoints fused? If yes, how to solve the inconsistency of the detection results of images from different viewpoints in the overlap area?

---

### Official Review · Reviewer_JxFc · 2023-10-29

**Soundness:** 2 fair
**Presentation:** 3 good
**Contribution:** 3 good
**Rating:** 5
**Confidence:** 5

**Summary:**

This work studies three design choices within the late-fusion framework for Long-Tailed 3D Detection (LT3D). The authors present a simple late-fusion approach that fuses 2D RGB-based detections and 3D LiDAR-based detections and achieves state-of-the-art results on LT3D benchmarks.

**Strengths:**

- Instead of fusing the RGB 3D detector with the LiDAR 3D detector, this work proposes using an RGB 2D detector to calibrate 3D detection classification and 3D score to help LT3D and achieve new SotA performance.
- Extensive analyses of different choices of matching 2D and 3D, score calibration and different 2D and 3D detectors.

**Weaknesses:**

- Although the performance is quite good. It seems like this work is more like a good engineering work. The novelty is limited.
- The 2D RGB detector works more like just a classifier. It looks like it is missing the most important baseline, e.g., using a strong classification network for the projected 3D boxes and still being able to perform semantic and score calibration.

**Questions:**

- Datasets like nuScenes [1] and Argoverse 2 [2] have multiple ring cameras. How do you deal with the projected 3D boxes divided into different camera scenarios?


[1] Holger Caesar, Varun Bankiti, Alex H Lang, Sourabh Vora, Venice Erin Liong, Qiang Xu, Anush Krishnan, Yu Pan, Giancarlo Baldan, and Oscar Beijbom. nuscenes: A multimodal dataset for autonomous driving. In Proceedings of the IEEE/CVF Conference on Computer Vision and Pattern Recognition, 2020.

[2] Benjamin Wilson, William Qi, Tanmay Agarwal, John Lambert, Jagjeet Singh, Siddhesh Khandelwal, Bowen Pan, Ratnesh Kumar, Andrew Hartnett, Jhony Kaesemodel Pontes, et al. Argoverse 2: Next generation datasets for self-driving perception and forecasting. arXiv preprint arXiv:2301.00493, 2023.